# Transition to bio-based plastic packaging reveals complex climate–biodiversity trade-offs

Bilal Erradhouani [1,2], Veronique Coma[2], Guido Sonnemann [1] & Philippe Loubet [1] ✉

Plastics are a major contributor to global greenhouse gas emissions and biodiversity loss, with packaging accounting for around 40% of European plastic production. Bio-based plastics are often promoted as a climate-friendly alternative, yet their broader environmental implications remain unclear. Here, we conduct a harmonized life cycle assessment of fossil- and bio-based plastic packaging, integrating end-of-life fate and plastic leakage. We find that while bio-based plastics reduce greenhouse gas emissions, they increase ecosystem damage, primarily driven by land use. At the product level, outcomes are sensitive to feedstock origin and waste management. When mismanaged, environmentally persistent bio-based plastics contribute substantially to ecosystem damage. Scaling scenarios for Europe show that even complete substitution with bio-based plastics cannot offset the environmental burden of continued demand growth. Only strong demand-side measures, particularly demand reduction and improved circularity, can mitigate trade-offs across climate change and biodiversity, calling for a shift from material substitution to systemic interventions in production and consumption.

Plastics have become integral to modern economies, driven by their durability, low weight, and versatility. Global production reached 460 million tons in 2019, bringing the total to over 9.5 billion tons since 1950[1,2]. Notably, 80% of all plastics ever produced were made in the past two decades, reflecting the exponential rise in global demand[3]. Initially developed for durable applications, plastics are now predominantly used in disposable or short-lived products. In EU-27 + 3 (European Union plus Norway, Switzerland, and the United Kingdom), packaging alone accounts for 40% of plastic production, with 60% dedicated to food and beverage applications[4–6].

While plastic packaging plays a critical role in preserving goods and preventing food waste[7], its environmental cost is escalating. The sector emits around 2 Gt $CO_2$-equivalent ($CO_2e$) annually, accounting for approximately 4.5% of global greenhouse gas (GHG) emissions, mainly due to its reliance on fossil-based feedstocks and energy-

intensive production processes[8]. Simultaneously, the accumulation of plastic waste has triggered a global pollution crisis[9]. An estimated 8 million tonnes of plastic enter the oceans each year[10], where they persist, break down into microplastics (MPs), and accumulate in marine food webs[11,12]. Plastic debris harms wildlife through entanglement and ingestion, introduces toxic additives into ecosystems, and may pose risks to human health via contamination of air, water, and food[13,14]. Failures in end-of-life management, particularly in the packaging sector, remain a major source of uncontrolled emissions to terrestrial and aquatic environments[5,15].

In response, bio-based plastics, deriving from biomass, have gained attention as substitutes for fossil-based materials[16]. Broadly, these can be divided into "drop-in" bio-based plastics (e.g., bio-based polyethylene), which are chemically identical to their fossil-based counterparts and compatible with existing recycling systems; and

[1]Université de Bordeaux, CNRS, Bordeaux INP, ISM, UMR 5255, Talence, France. [2]Université de Bordeaux, CNRS, Bordeaux INP, LCPO, UMR 5629, Pessac, France. ✉e-mail: philippe.loubet@u-bordeaux.fr

compostable bio-based plastics (e.g., polylactic acid), which are designed to biodegrade under industrial composting conditions and thus require distinct waste management infrastructure. By capturing biogenic $CO_2e$ during plant growth, these materials offer the prospect of lowering life cycle carbon emissions, and are increasingly promoted as part of circular bioeconomy strategies[17]. Although bio-based plastics currently represent only 1–2% of global production, this share is expected to grow[18]. Substituting two-thirds of conventional plastics with bio-based alternatives could avoid 241–316 $MtCO_2e$ annually[19]. Such transitions are encouraged by regulatory instruments like the Single-Use Plastics Directive[20].

However, the large-scale deployment of bio-based plastics faces significant barriers. High production costs make them less competitive in price-sensitive markets, and the lack of standardized recycling or composting infrastructure for the more innovative materials hinders their integration into existing waste systems[21]. More fundamentally, their sustainability remains debated. First-generation feedstocks, typically derived from readily fermentable sugars in edible polysaccharides such as corn and sugarcane, demand substantial land, water, and fertilizer inputs. This raises concerns about competition between food, fuel, and plastics, as well as broader ecological degradation[22]. Second-generation biomass, derived from non-edible biological waste, has emerged in recent years as a more ethically viable and widely available feedstock, albeit a more complex one. Beyond the technical challenges limiting large-scale deployment, second-generation feedstocks may still induce indirect land-use change when their sourcing competes with existing land functions or displaces other uses. Furthermore, despite their renewable origin, mismanaged bio-based plastics can persist, degrade and fragment in the environment much like their fossil-based counterparts, thereby contributing to plastic pollution.

Although life cycle assessment (LCA) has become a standard tool for evaluating the environmental performance of plastics, its application has often been limited in scope, with most studies emphasizing climate-related metrics[23,24]. Furthermore, few incorporate harmonized inventory data across life cycle stages or account for plastic leakage into the environment[25]. Large-scale assessments have advanced understanding of the GHG implications of plastic production and disposal, but they still overlook broader environmental dimensions, including impacts on ecosystem quality and human health[26]. Recently, analyses examining the substitution of conventional plastics with bio-based alternatives have provided a robust understanding of the implications of plastic pollution and carbon mitigation strategies for cropland expansion[27]. However, these studies do not integrate other environmental dimensions within a comprehensive life cycle framework. As bio-based plastics gain market share, a more comprehensive understanding of their systemic impacts is urgently needed[28].

Here, we conduct a harmonized LCA of fossil- and bio-based plastic packaging to quantify environmental trade-offs across climate change, ecosystem quality, and human health. We include case studies of five bio-based plastics: bio-based polyethylene (bio-PE), bio-based polyethylene terephthalate (bio-PET), polylactic acid (PLA), polyhydroxybutyrate (PHB), and thermoplastic starch (TPS); and seven fossil-based plastics: polyethylene terephthalate (PET), high-density polyethylene (HDPE), low-density polyethylene (LDPE), polypropylene (PP), polyvinyl chloride (PVC), polystyrene (PS), and polyurethane (PUR).

The analysis is structured at three levels. At the material level, we examine life cycle stage contributions and perform a detailed trade-off analysis across midpoint and endpoint indicators. At the product level, we compare packaging providing equivalent functionality, accounting for variability in feedstock origin and end-of-life treatment. At the macro level, we explore large-scale substitution scenarios for Europe under alternative demand and energy trajectories. The study adopts an attributional perspective, and thus no consequential effects such as

indirect land-use change are considered. The scope includes first- and second-generation bio-based plastics but excludes third-generation feedstocks due to the lack of reliable production data. All life cycle stages are included, with explicit integration of plastic leakage quantification and its associated impact assessment.

## Results and discussion
### Bio-based plastics reduce carbon footprint at the cost of increased ecosystem damage from land use

Figure 1 presents the impact results associated with the production of market-representative packaging materials from primary plastics, excluding the incorporation of recycled content. Although the uncertainty range is wider for bio-based polymers, reflecting the higher variability of agricultural and industrial inventories, their GHG emissions remain generally lower than those of fossil-based counterparts. The partial overlap of uncertainty ranges, observed mainly for bio-PET, bio-PE, and PHB, indicates that this advantage may narrow under less favorable production conditions but remains robust on average (Fig. 1a). This benefit is mainly due to $CO_2$ uptake during plant growth, which serves as the feedstock for biopolymer production. Net emissions range from 3.5 to 5.5 $kgCO_2e$ $kg^{-1}$ for bio-based packaging, compared to 5 to 9.5 $kgCO_2e$ $kg^{-1}$ for fossil-based equivalents. In both material types, the polymer production stage dominates the carbon footprint. For bio-based systems, this is due to the energy-intensive conversion of biomass into fermentable carbohydrates and polymer precursors; for fossil-based systems, it is linked to steam cracking of hydrocarbon feedstocks. In contrast, the raw-material extraction phase, whether fossil or agricultural, contributes relatively little to overall climate impacts (Fig. 1f). End-of-life scenarios also influence carbon outcomes. Fossil-based plastics stored in landfills tend to sequester carbon over a 100-year time horizon, whereas biodegradable plastics partially degrade, releasing biogenic $CO_2$ and $CH_4$[29]. Composting, when available, yields primarily $CO_2$ emissions with limited methane, while landfilling under anaerobic conditions results in a split of $CO_2$ and $CH_4$ emissions[30]. The net benefit of bio-based packaging in terms of climate change, therefore, depends on both production inputs and end-of-life conditions.

However, this carbon advantage comes with a significant trade-off in ecosystem quality. Impacts expressed in PDF $m^2$ yr (potentially disappeared fraction of species over a given area and time) are consistently higher for bio-based materials, ranging from 1 to 5 PDF $m^2$ yr $kg^{-1}$, compared to 0.3 to 2 PDF $m^2$ yr $kg^{-1}$ for fossil-based options (Fig. 1c). These damages are mainly driven by land use, which includes occupation and transformation (Methods section), followed by acidification (both terrestrial and marine), as shown in Fig. 2. The magnitude of land-use impact from bio-based plastics is linked to the nature of the feedstock and conversion efficiency. For instance, producing bio-PE, which requires carbon-rich ethylene, demands approximately 6 kg of corn kg per kg of polymer, nearly three times more biomass than PLA production. As a result, land occupation and the associated pressures on biodiversity are amplified. First-generation PLA packaging relies on agricultural land, exerting considerable stress on ecosystems in the short-term (Fig. 2b). Land occupation and transformation account for roughly 60% of the total impact for corn-based PLA and sugarcane-based PHB. However, these impacts are partially offset by reduced climate change effects due to biogenic carbon sequestration. Shifting to second-generation residual biomass for PLA production reduces the land-use contribution to ecosystem damage. This decrease is explained by the application of economic allocation to distribute the environmental burden among the outputs of corn cultivation. A lower economic value is attributed to corn stover. The environmental benefit of moving from first- to second-generation feedstocks hinges on the assumption that these residues are treated as biowastes. If co-products from first-generation crops acquire greater economic value, a larger share of land-use burden

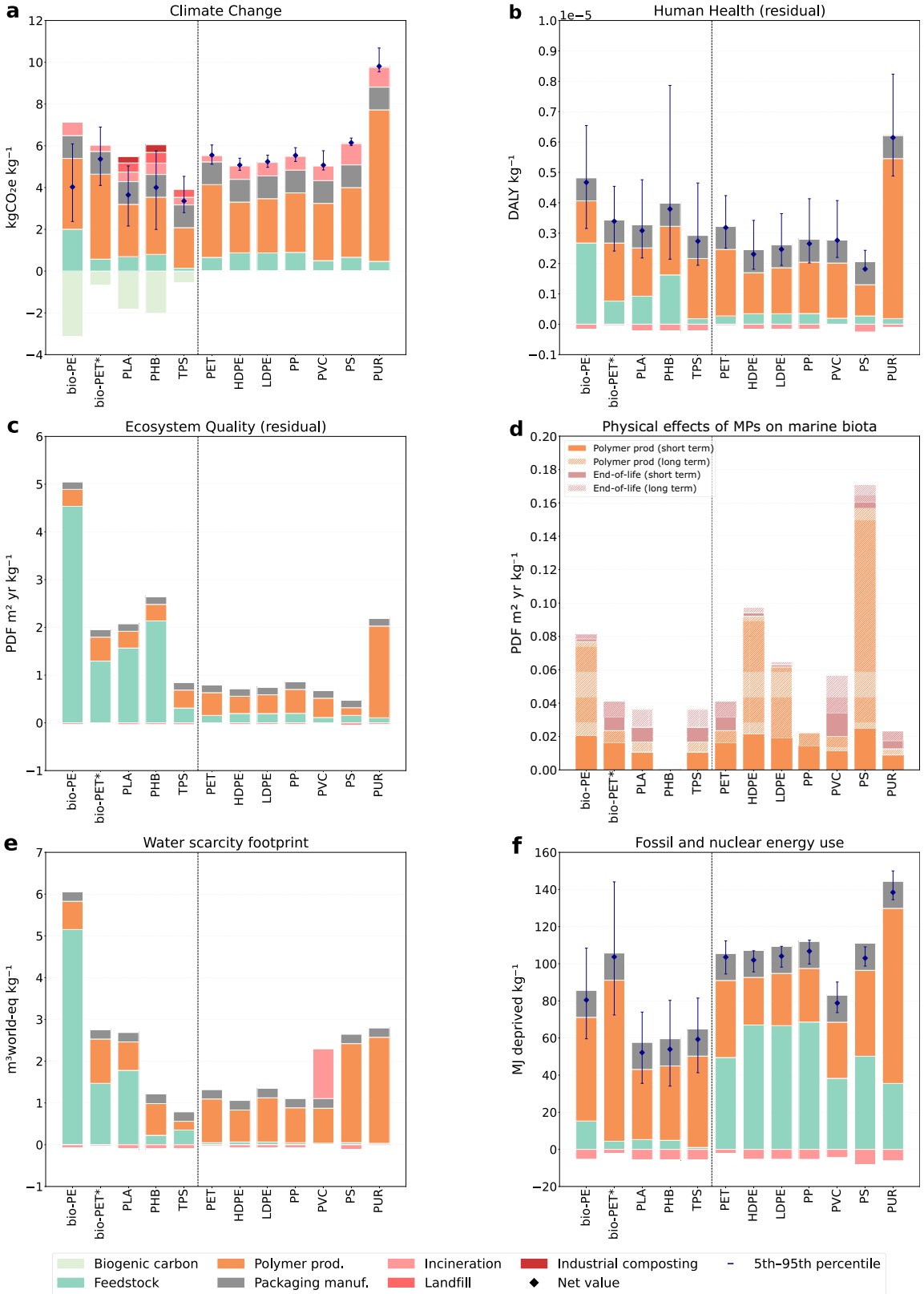

would be allocated to them. Moreover, some second-generation resources can also be derived from dedicated cellulose crops, such as wood, which may in turn require significant land areas. For fossil-based packaging, climate change emerges as the primary driver of ecosystem damage. The low contribution of land use reduces the overall impact (Fig. 2b).

Another emerging concern is the impact of plastic leakage into the environment, which remains underrepresented in traditional LCA models[31]. Based on average loss rates in Europe, the production of 1 kg of packaging results in emissions of approximately 0.1 g of MPs during polymer production and an additional 10 g of macroplastic debris at end-of-life[32,33]. Using recent characterization methods, we

**Fig. 1 | Environmental impacts of market-representative bio- and fossil-based primary plastic packaging.** Bio-based packaging includes bio-polyethylene (bio-PE), bio-polyethylene terephthalate containing 30% bio-based materials (bio-PET*), polylactic acid (PLA), polyhydroxybutyrate (PHB), and thermoplastic starch (TPS); fossil-based packaging includes polyethylene terephthalate (PET), high-density polyethylene (HDPE), low-density polyethylene (LDPE), polypropylene (PP), poly-vinyl chloride (PVC), polystyrene (PS), and polyurethane (PUR). All impacts are assessed with IMPACT World+ 2.0.1 (footprint version) except (**d**). *Residual* refers to damage to ecosystem quality and human health, excluding the contribution of

climate change and water availability. In (**a**, **b**, **f**), error bars represent the 5th–95th percentile from Monte Carlo simulations based on inventory data uncertainty. **a** Greenhouse gas emissions. **b** Damage to human health (residual) expressed in disability-adjusted life years (DALY). **c** Damage to ecosystem quality (residual) expressed as the potentially disappeared fraction of species over a given area and time (PDF m² yr). **d** Physical effects of microplastics (MPs) on marine biota associated with plastic leakages during polymer production (0.01% loss) and end-of-life mismanagement (1% loss). **e** Water scarcity footprint. **f** Fossil and nuclear energy use.

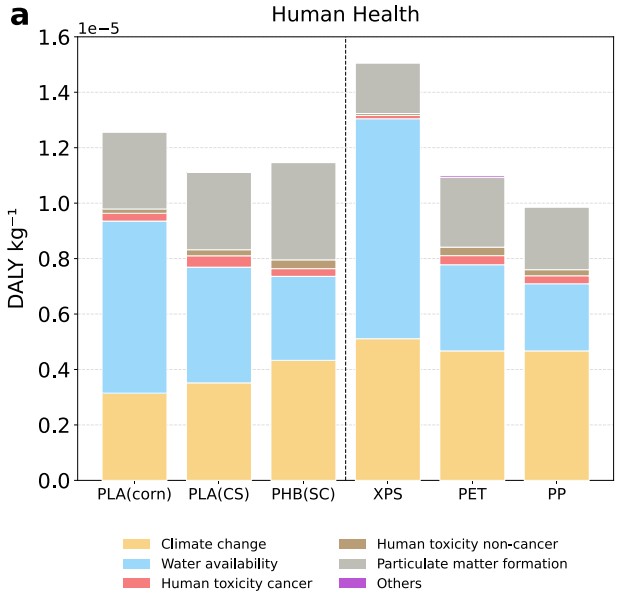

**Fig. 2 | Contributions of midpoint impact categories to short-term (≤100 years) damage on human health and ecosystem quality from the production of primary bio- and fossil-based packaging.** Comparison between polylactic acid (PLA) from corn and corn stover (CS), polyhydroxybutyrate (PHB) from sugarcane (SC), and three fossil-based references: extruded polystyrene (XPS), polyethylene terephthalate (PET), and polypropylene (PP). All impacts are assessed with IMPACT World+ 2.0.1 (expert version). **a** Contributions of midpoint indicators to short-term

damage on human health, expressed in disability-adjusted life years (DALY). **b** Contributions of midpoint indicators to short-term damage on ecosystem quality, expressed as the potentially disappeared fraction of species over a given area and time (PDF m² yr). The physical effects of microplastics (MPs) on marine biota were included based on two loss scenarios reflecting the economic level of the country responsible for packaging end-of-life management.

estimated the physical impact of these losses on marine ecosystems in the short-term (≤100 years) and long-term (>100 years)[34–36]. An estimation of the uncertainties associated with the characterization factors (CFs) used is provided in Supplementary Fig. 2. We considered similar effect factors to characterize the impact of different microplastic emissions on marine biota, making damage comparisons between polymers only dependent on the residence time of debris[34]. Primary microplastics (MP₁), which are directly emitted from the technosphere during pellet production, and secondary microplastics (MP₂), arising from the fragmentation of macroplastics reaching the environment, exert comparable contributions under the low short-term fragmentation rate assumed for packaging end-of-life (Fig. 1d). Polymers with low degradation rates in the marine environment exhibit longer residence times, increasing exposure risks for marine organisms. For infinite-term impacts (sum of the short and long term), all non-biodegradable polymers in the marine environment cause damage ranging from 0.02 to 0.17 PDF m₂ yr kg ₋₁ of packaging produced. Packaging debris composed of polymers with high degradation rates, such as PHB, exhibits minimal persistence in the marine environment, resulting in significantly attenuated effects, with impacts approaching 0 PDF m² yr kg⁻¹. Under a scenario of average fragmentation of macroplastics released

post-consumption[25], the results reveal a minor contribution to the overall impacts across the different life cycle stages in high-income countries with effective waste collection systems (Fig. 2b). However, this contribution becomes significant in low-income countries, where end-of-life plastic management is challenging. In such contexts, the impact of PLA packaging leakages can account for up to 30% of the total ecosystem burden.

Human health damages from plastic packaging result from multiple pathways, including particulate matter, water availability, and human toxicity (Fig. 2a). While these are covered in life cycle models, uncertainties persist, especially for toxicity-related impacts, due to data gaps and simplified modeling[37]. Direct human exposure to microplastics or chemical additives during the use or leakage phases is not included in conventional LCAs, as no validated human-health exposure and effect characterization models currently exist for plastics. The results indicate a disadvantage for bio-based packaging mainly due to the fertilizers use and the substantial water requirements associated with crop cultivation (Figs. 1b and 2b). Bio-based packaging production leads to an impact ranging from 3 × 10⁻⁶ to 5 × 10⁻⁶ Disability-Adjusted Life Years (DALY) kg⁻¹, corresponding to a loss of approximately two to three minutes of life lost in total when aggregated across the population (Fig. 1b). Excluding PUR, the production of fossil-based packaging

results in an impact ranging from $0.2 \times 10^{-5}$ and $0.3 \times 10^{-5}$ DALY $kg^{-1}$ (about one to two minutes of life lost).

## Bio-based product-level impacts are highly sensitive to feedstock and end-of-life

To assess how upstream and downstream choices affect environmental performance at the product level, we analyzed the life cycle of a PLA food tray (13.9 g). The functional unit was defined as the ability to contain 1 liter of food product, ensuring comparability across materials with different densities and compositions[38]. We examined the effects of feedstock type (first- or second-generation), recycled content incorporation, and end-of-life options for PLA packaging, in comparison with fossil-based trays, including PET (18.5 g), PP (11.9 g), and extruded polystyrene (XPS) (9.9 g).

Feedstock origin has a major influence on ecosystem impacts, and a more limited but non-negligible influence on climate. Switching between first-generation feedstocks (corn and sugarcane) has only minor effects on GHG emissions (18 vs. 14 $gCO_2e$ $tray^{-1}$), due to similar agricultural practices and allocation rules (Fig. 3a). In contrast, the use of residues such as corn stover reduces agricultural emissions, but leads to higher impacts during conversion, which is more energy-intensive and less efficient. As a result, the total climate impact of stover-based PLA was less favorable than expected (20 $gCO_2e$ $tray^{-1}$), highlighting the trade-off between benefits during the agricultural phase, and burdens during manufacture. When excluding the benefit of biogenic $CO_2$ uptake, PLA production from recycled pellets results in lower climate change impacts (26 $gCO_2e$ $tray^{-1}$) compared to production from virgin feedstock. From an ecosystem quality perspective, however, the differences were substantial: stover-based PLA tray has a footprint of 0.015 PDF $m^2$ yr, far lower than the 0.027 and 0.031 associated with corn- and sugarcane-based trays (Fig. 3b). These results confirm that land use is the dominant variable in ecosystem damage and that residue-based systems offer real advantages. Geographic context also influences impact levels. For example, sugarcane cultivation in Brazil for PHB production benefits from natural rainfall, resulting in a lower water footprint than artificially irrigated crops in other regions (Fig. 1e). By reducing input requirements and eliminating the need for agricultural cultivation, the production of PLA from recycled pellets significantly lowers ecosystem damage (-0.003 PDF $m^2$ yr $tray^{-1}$).

Over a 100-year time horizon, end-of-life pathways can produce contrasting effects on climate change and ecosystem outcomes (Fig. 3c and 3d). Landfilling and incineration with energy recovery result in comparable GHG emissions (15 $gCO_2e$ $tray^{-1}$), as carbon is either partly sequestered in landfills or oxidized to $CO_2$ during combustion, with the latter offset by energy recovery. Composting leads to higher emissions (26 $gCO_2e$ $tray^{-1}$), since PLA undergoes extensive aerobic degradation, converting most of its biogenic carbon into $CO_2$. Although the compost generated can substitute for soil amendments, the associated benefit remains minor and does not offset emissions to the same extent as energy recovery. Recycling results in the lowest GHG emissions, because the polymer carbon remains stored in the recycled material, delaying its release. When PLA is littered, a small amount of $CO_2$ is emitted due to its slow biodegradation under marine conditions, but the main burden shifts to ecosystem quality, as persistent microplastics affect marine biota (0.012 PDF $m^2$ yr $tray^{-1}$). This level of impact far exceeds that of any controlled treatment, underscoring the critical role of preventing plastic leakage for ecosystem.

Comparisons with fossil-based trays reveal contrasting trends across impact indicators. Figures 3e and 3f presents the results for climate change and ecosystem quality impacts, respectively, combining different scenarios for raw-material sourcing and end-of-life management. The baseline PLA production setup, representing average market conditions, is complemented by lower- and upper-bound

estimates (*PLA-low* and *PLA-high*) that capture geographical variability in feedstock yield and energy intensity. Assumptions underlying these regionalized cases are provided in Supplementary Table 14.

In terms of greenhouse gas emissions, PLA trays consistently outperform those made from PET (110 $gCO_2e$ $tray^{-1}$) and PP (75 $gCO_2e$ $tray^{-1}$) across all configurations. Under a mixed end-of-life scenario, PLA shows climate performances comparable to XPS (50–60 $gCO_2e$ $tray^{-1}$) in the baseline case and substantially lower in the lower-bound case. In contrast, under the upper-bound conditions, only complete recycling at end-of-life would keep emissions below those of XPS.

For ecosystem quality, feedstock type exerts a stronger influence than geographic variability. Differences between *PLA-low* and *PLA-high* are smaller than those between first-generation, residue-based, and recycled systems. Only trays made from recycled pellets or corn stover (*low* scenario), and managed properly at end-of-life, achieve results within the fossil-based range (0.005–0.015 PDF $m^2$ yr $tray^{-1}$).

## Scaling bio-based plastics reveals that only strong demand reduction avoids ecological trade-offs

To assess the system-wide consequences of bioplastic transitions, we modeled the environmental impacts of plastic packaging demand in Europe from 2020 to 2050 under various substitution scenarios (Fig. 4). We simulated a progressive replacement of fossil-based packaging with first-generation bio-based alternatives, under two energy contexts (current mix vs. full decarbonization), and three annual growth trajectories for packaging demand: −3%, 0% (no growth) and 3%. In all cases, substitution was modeled as a gradual transition toward 100% bio-based packaging by 2050, thereby isolating the effect of substitution from other structural changes. In addition, gradual increases in recycling rates were included in the analysis to account for the potential incorporation of recycled materials in packaging production.

Assuming that all sectors align with the GHG emission reduction trajectory recommended by the Intergovernmental Panel on Climate Change (IPCC)[39], the target would be to reach approximately 17 $MtCO_2e$ by 2050 (Fig. 4a). If packaging demand increases by 3% annually, approximately the current European growth rate[33], total GHG emissions will more than double by 2050 under a business-as-usual scenario, reaching almost 250 $MtCO_2e$. Full substitution with bio-based plastics could prevent this increase, containing emissions at around 100 $MtCO_2e$ under a decarbonized system incorporating recycled materials (*Bio-low*). Nevertheless, even under the most favourable trajectory, emission levels would remain fivefold higher than the target recommended by the IPCC. In a no-growth scenario, replacing fossil plastics with bio-based alternatives results in a significant decrease in carbon emissions, potentially reaching 50 $MtCO_2e$ by 2050. Yet this level would still remain clearly above the threshold recommended by the IPCC. Only a reduction in packaging demand of at least 3% $year^{-1}$ would substantially lower $CO_2e$ emissions to levels aligned with IPCC recommendations, potentially positioning Europe as a key player in global climate change mitigation.

In contrast, ecosystem damage increases significantly under all substitution scenarios (Fig. 4b). Replacing fossil-based plastics with first-generation bio-based alternatives could lead to a three- to fivefold increase in cumulative ecosystem damage between 2020 and 2050. This is primarily due to the expansion of agricultural land required to supply biomass feedstocks. In the 3% growth case scenario with full substitution without incorporation of recycled material (Bio-Max), the total impact reaches $1.10 \times 10^{11}$ PDF $m^2$ yr. This metric represents the overall potential loss of species, integrating all stressors affecting ecosystem quality at the endpoint level, with land occupation and transformation being the dominant contributors. To illustrate the order of magnitude, this value would correspond to a level of biodiversity damage comparable to that expected over roughly 10% of Europe's current arable land, acknowledging that most of the loss

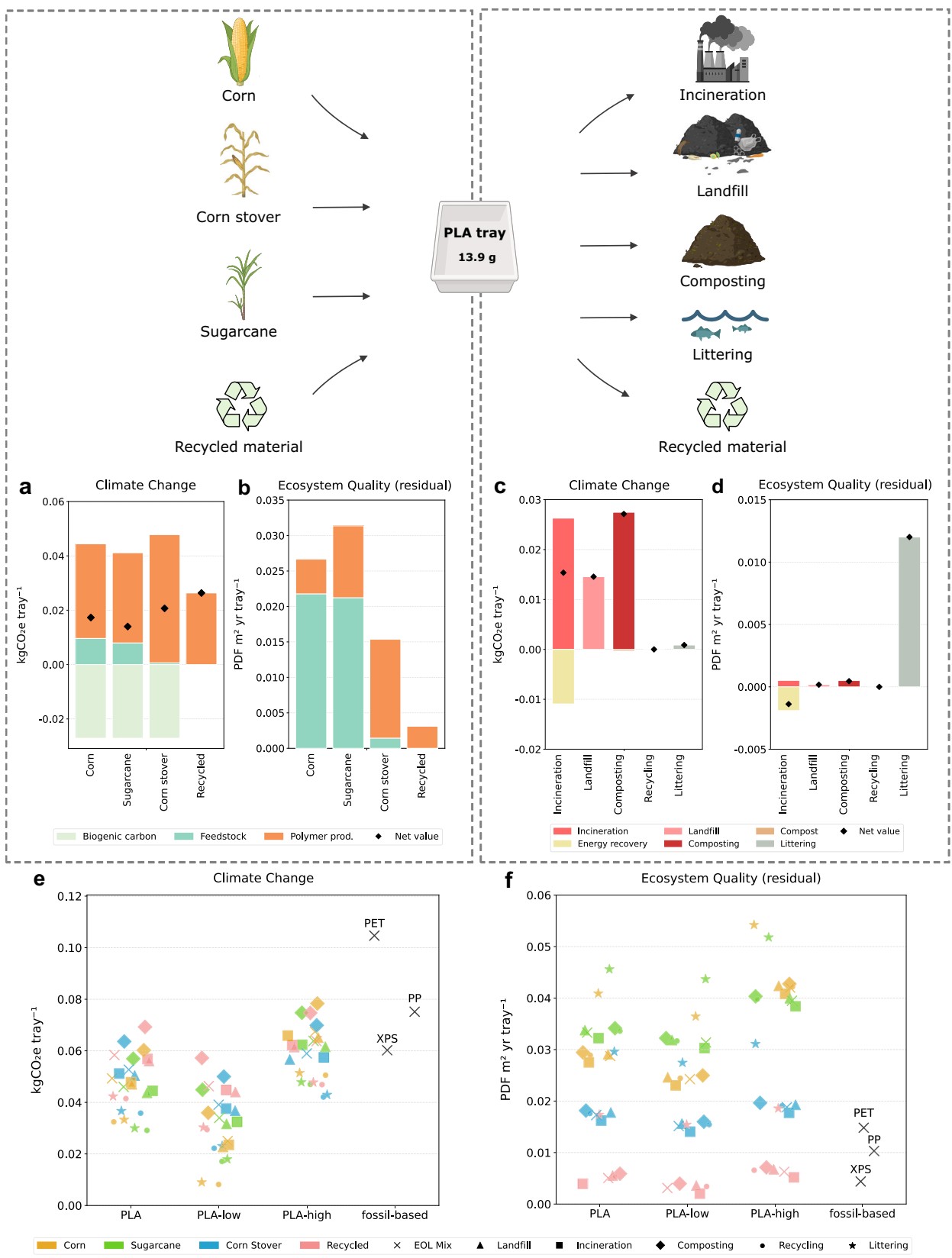

would result from agricultural expansion. While this comparison remains illustrative, given the very low likelihood of achieving full substitution to bio-based plastics and the fact that a substantial share of biomass production would occur outside Europe, it highlights the potential ecological burden associated with large-scale substitution. Even in the most favorable energy scenario, ecosystem pressures remain significantly higher under bio-based packaging than under continued fossil use, revealing a strong climate–biodiversity trade-off at scale.

For both analyzed indicator, the gap between fossil-based and bio-based trajectories narrows significantly with decreasing packaging demand, underscoring the critical importance of consumption

**Fig. 3 | Environmental trade-offs of polylactic acid (PLA) packaging across feedstock, end-of-life pathways, and fossil-based alternatives.** All impacts are assessed with IMPACT World+ 2.0.1 (footprint version). The physical effects of microplastics (MPs) on marine biota have been added to the ecosystem quality assessment, expressed as the potentially disappeared fraction of species over a given area and time (PDF m² yr). *Residual* refers to damage to ecosystem quality, excluding the contribution of climate change and water availability.
**a**, **b** Greenhouse gas (GHG) emissions (**a**) and damage to ecosystem quality (residual) (**b**) associated with feedstock and polymer production stages of PLA trays

derived from corn, corn stover, sugarcane, and recycled PLA. **c**, **d** GHG emissions (**c**) and damage to ecosystem quality (residual) (**d**) generated across end-of-life scenarios: incineration with energy recovery, landfill, composting with nutrient recovery from compost, recycling (burden-free), and littering (short-term impacts on marine biota). **e**, **f** GHG emissions (**e**) and damage to ecosystem quality (residual) (**f**) for PLA and fossil-based trays−extruded polystyrene (XPS), polyethylene terephthalate (PET), and polypropylene (PP)−across combined feedstock and end-of-life scenarios. Created in BioRender. Erradhouani, B. (2026) https://BioRender.com/suv9dat.

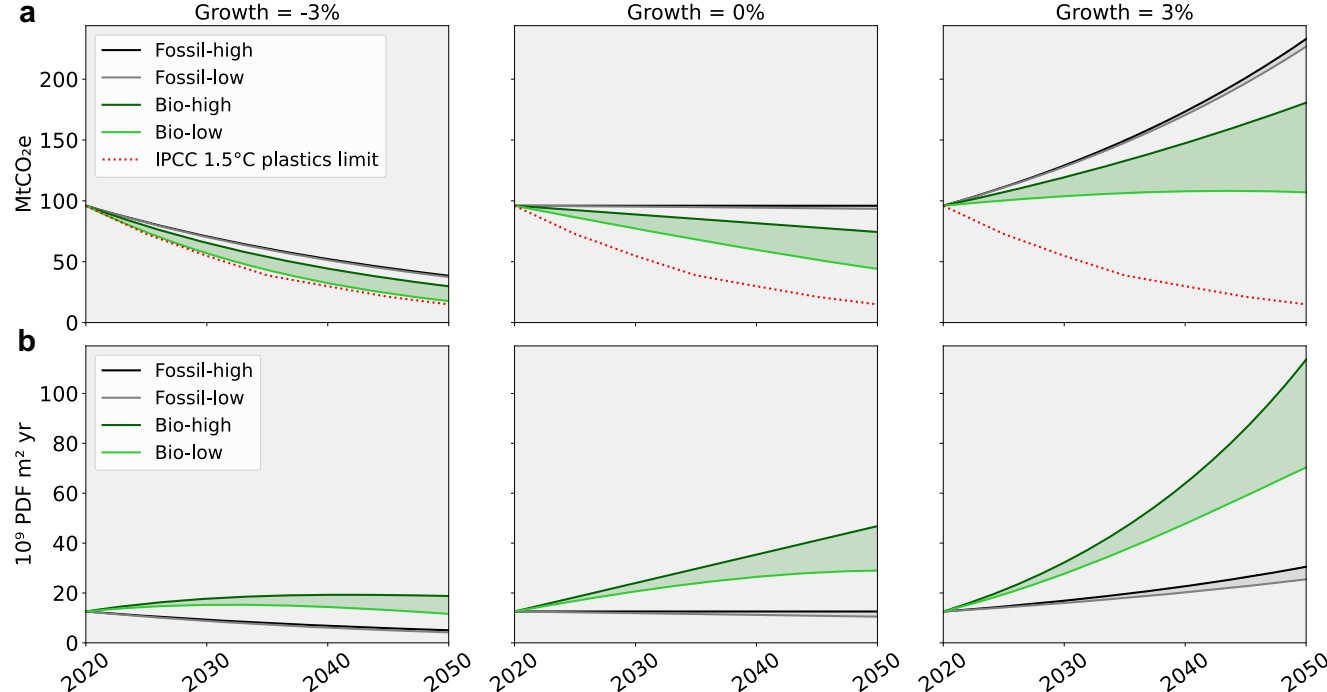

**Fig. 4 | Impacts of packaging plastic demand in Europe under different scenarios of substituting conventional materials with bio-based alternatives.** The *Fossil-high* pathway represents a business-as-usual scenario, *Fossil-low* includes the incorporation of recycled polymers, *Bio-high* involves a gradual substitution of fossil-based polymers with bio-based alternatives, and *Bio-low* further incorporates progressive energy mix decarbonization alongside recycled polymer incorporation. Scenarios are evaluated under three demand trajectories: demand reduction

(−3% yr⁻¹), no growth (0%), and demand growth (+3% yr⁻¹). All impacts are assessed with IMPACT World+ 2.0.1 (footprint version). **a** Total greenhouse gas emissions. The red dashed line represents an indicative 1.5 °C-aligned greenhouse gas emissions pathway for the plastics sector, derived from IPCC mitigation scenarios.
**b** Total damage to ecosystem quality, expressed as the potentially disappeared fraction of species over a given area and time (PDF m² yr) and excluding the contribution of climate change and water availability.

reduction. Demand growth emerges as the dominant driver of environmental burden. While material substitution and energy decarbonization help reduce per-unit impacts, they are insufficient to achieve absolute reductions in a context of continued consumption growth. Our results show that, even under the most ambitious mitigation pathways, increased throughput of materials outweighs the gains from cleaner technologies. Substitution policies thus emerge as secondary levers that can help alleviate environmental pressure. Without curbing the structural growth in plastic demand, technological improvements and feedstock substitutions are likely to be offset. This highlights the need to couple substitution strategies with broader demand-side policies, material sufficiency measures, and shifts in consumption patterns.

### Rethinking sustainability in packaging system transitions beyond substitution
Our findings underscore the need to move beyond substitution alone. Demand-side strategies, including reduction, reuse, and improved end-of-life management, are essential to align packaging transitions

with planetary boundaries. While bio-based materials offer a promising pathway to decarbonize the plastics sector, our results reveal that this benefit comes at the cost of significant impact shifts towards ecosystem quality.

Replacing fossil-based polymers with bio-based alternatives can lead to unintended consequences, especially when relying on first-generation feedstocks that intensify land-use pressures and exacerbate biodiversity loss. Emerging second- and third-generation feedstocks, such as lignocellulosic residues, algae, or microbial biomass, could decouple bio-based plastics production from food systems and reduce land occupation. However, the scalability, availability, and cost competitiveness of these resources remain poorly understood. Second-generation biomass is constrained by the accessibility of agricultural residues (for example, straw, which is already in demand for applications such as animal bedding) and must avoid driving the exploitation of new areas, particularly forests. Third-generation sources, based on microalgae or microbial cultures, may further alleviate land-use pressures but remain constrained by high production costs, energy-intensive processing, and limited technological maturity[40].

Technical barriers, particularly in biomass conversion and polymer processing, limit the current performance of next-generation bio-based plastics and often increase energy demands. Substantial innovation in biopolymer design and processing efficiency is needed to realize their full potential while avoiding climate–biodiversity trade-offs[17].

Strategies focused on circularity can mitigate some of these pressures[41]. Increasing recycling rates and enabling reuse systems across both fossil- and bio-based packaging could substantially reduce life cycle impacts. For instance, enhancing recycling from current global rates (~23%) to theoretical maxima (~94%) could reduce planetary pressures by 10–49% for fossil plastics and 33–83% for bio-based plastics[42]. Yet, recycling remains constrained by infrastructure, material complexity, and contamination. Investments in sorting technologies, depolymerization processes, and harmonized standards are needed to unlock circular value chains[43]. Likewise, reusable packaging systems (including deposit-return schemes and refillable models) offer high mitigation potential, provided they overcome logistical hurdles and behavioral barriers. Such systems must be tailored to specific use cases to avoid burden-shifting and ensure net environmental benefits[44].

Ultimately, reducing the demand for single-use packaging is indispensable. Waste management improvements alone cannot compensate for rising plastic production. Policy instruments such as the European Union's Single-Use Plastics Directive have shown that regulatory action can effectively curb consumption. Moving forward, setting binding targets on virgin plastic production, stimulating material innovation through eco-design, and supporting behavioral change through fiscal and informational tools will be key levers[45].

Rethinking packaging sustainability thus requires an integrated approach that combines material innovation, circular economy principles, and ambitious demand-side interventions. This triad is essential not only to reduce the carbon footprint of plastics, but also to protect ecosystems and safeguard critical resources in a resource-constrained future. Mitigation measures must be accompanied by methodological advances, as scientific progress remains crucial for understanding contamination phenomena and for the quantified assessment of their impacts.

## Methods

### Life cycle assessment goal and scope
The objective of this study is to assess the burden shifting associated with carbon footprint reduction strategies for the plastic packaging industry. The selection of bio-based and fossil-based polymers reflects the composition of the current packaging market, following the approach of Zheng et al.[46]. Bio-based packaging materials facing substantial technical and economic barriers to large-scale adoption were excluded from the study scope due to uncertainties surrounding their feasibility as substitutes for conventional packaging. Similarly, cardboard and glass packaging were not included in the study due to the significant differences in their properties and usage compared to fossil-based plastic packaging.

A bottom-up approach was adopted to compare the environmental impacts associated with the production of fossil-based and bio-based packaging at different levels. At the material level, the functional unit (FU) is 1 kg of packaging. The assessment allows for the identification of burden shifting between fossil and biomass-based production while analyzing the contributions of different life cycle stages and the contribution of midpoint impacts on damage (Figs. 1 and 2). At the product level, the FU is a tray with a volume of 1 L. The assessment enables the study of technical and management strategies aimed at optimizing biomass use as an alternative (Fig. 3). At the macro level, the FU is the total European packaging demand. The assessment provides insights into the ecosystem effects of substituting conventional packaging with bio-based alternatives (Fig. 4).

### Harmonization procedure for the life cycle inventory
A harmonized inventory review of exchanged flows across the life cycle covered raw-material extraction, polymer production, packaging manufacturing, and end-of-life, ensuring result comparability and consistency (Supplementary Fig. 1). The conversion and end-of-life rates are provided in Supplementary Table 1.

Two harmonization layers were applied: system harmonization (align FU, system boundaries, and methodological choices across reviewed LCAs) and technical harmonization (standardize inventories by adjusting data and calculations). The workflow was: (1) disaggregate life cycle inventory (LCI) data to unit-process level; (2) classify by life cycle stage and polymer; (3) sort by data quality; (4) complete gaps to match the FU. When multiple sources described the same process, mass/energy balance was verified before integration. For sources with consistent input shares, averages were used. Where ecoinvent v3.10 generic data existed, it was incorporated either as an average with literature values or as a reference for gap-filling (Supplementary Table 3). Because references flows varied (resin, bottles, film, cups, etc.), 1 kg of packaging was used as the standard FU.

Fossil-based packaging inventories rely on aggregated ecoinvent v3.10 data across life cycle stages. Foreground flows (chemicals, energy, infrastructure, waste) were compiled and the background (energy/material supplies) was taken from ecoinvent v3.10 global/market mixes. Given limited traceability, activity data for infrastructure, energy, and chemicals prioritize global market suppliers.

**Feedstock and carbon sequestration.** The environmental impacts of feedstock were modeled based on first-generation biomass production due to its market readiness. Corn cultivated in the United States was selected as the feedstock for PLA, bio-PE, bio-PET, and TPS production, while sugarcane grown in Brazil was used for PHB production. For systems based on second-generation feedstocks, the agricultural phase was allocated economically between grain and stover (95% and 5%, respectively) following the World Food LCA Database.

The $CO_2$ uptake during plant growth was calculated directly from the carbon content of each polymer and the biogenic fraction of feedstock used, ensuring a consistent carbon mass balance across life cycle stages. This method enables tracking of biogenic carbon through end-of-life processes and does not represent the total photosynthetic fixation by the plant, nor does it follow economic allocation rules. The resulting sequestration factors are 3.14 $kgCO_2$ $kg^{-1}$ for bio-PE, 0.69 for bio-PET, 1.83 for PLA, 2.04 for PHB, and 0.56 for TPS.

**Bio-based polymer production.** Bio-polyethylene: The production of bio-PE from corn begins with the enzymatic hydrolysis of starch, breaking it down into glucose monomers[30]. The glucose undergoes fermentation by yeast to yield bioethanol, which is then purified. In the final stage, bioethanol is subjected to catalytic dehydration, converting it into ethylene ($C_2H_4$), the key monomer for polyethylene synthesis (Supplementary Table 4).

Bio-polyethylene terephthalate: The production of bio-PET consists of the condensation polymerization of two chemicals: monoethylene glycol (MEG) and terephthalic acid (TPA), respectively accounting for 30% and 70% by weight[47–49]. While replacing fossil-based MEG with bio-based MEG is relatively widespread, PTA remains largely fossil-based. Bio-PET retains the same properties as traditional PET, allowing it to integrate seamlessly into existing recycling systems. Bioethanol obtained from sugar fermentation is dehydrated to ethylene, which is subsequently oxidized to ethylene oxide and hydrolyzed to produce bio-based monoethylene glycol (bio-MEG). In parallel, TPA is synthesized from the catalytic oxidation of para-xylene (p-xylene) with acetic acid (Supplementary Table 5).

Polylactic acid: PLA production from corn involves several key processes, which were integrated into a tailored life cycle inventory[50]. First, the dried corn grains undergo wet milling to separate their

components. The extracted starch is then hydrolyzed enzymatically into glucose monomers, which serve as the substrate for bacterial fermentation. To regulate pH during fermentation, calcium carbonate is introduced as a neutralizing agent, leading to the formation of calcium lactate upon reaction with calcium hydroxide. The solution is subsequently acidified with sulfuric acid to yield lactic acid, which undergoes purification before being converted into PLA through a ring-opening polymerization process (Supplementary Table 6). Details on PLA production from corn stover and sugarcane are provided in Supplementary Tables 7 and 8.

Polyhydroxybutyrate: PHB is a biodegradable polyester produced by bacteria as a carbon and energy storage compound[51]. Sucrose, the primary fermentable sugar, is extracted and refined from the juice before being supplied to PHB-producing bacteria. Under nutrient-limited conditions with excess carbon availability, these bacteria accumulate PHB in the form of intracellular granules. Following fermentation, the bacterial biomass is recovered through centrifugation or filtration. PHB is then extracted using solvent-based methods, followed by purification, drying, and processing into powder or granules for subsequent use (Supplementary Table 9).

Thermoplastic starch: TPS is a biodegradable polymer derived from corn starch. The production process begins with the milling of corn to separate starch from proteins, fibers, and oil. The extracted starch is then purified and dried into a fine powder. During processing, the starch is blended with plasticizers and subjected to heat and mechanical shear, leading to gelatinization, where the starch granules swell, absorb the plasticizers, and transition into a thermoplastic phase suitable for further processing. The inventory is based on aggregated data from the environmental product declaration of Mater-Bi (Ecoinvent 3.10).

**Packaging manufacturing.** Due to the variability in processing technologies, the manufacturing of bio-based and fossil-based polymer resins into packaging was modeled using the generic extrusion and thermoforming processes provided by ecoinvent, based on global market data. Differences in manufacturing processes between conventional and bio-based packaging alternatives were not accounted for, as inventory data specific to these variations remain challenging to obtain.

**Greenhouse gas emissions from incineration, landfill, composting, and littering.** GHG emissions from different end-of-life scenarios were estimated using different modeling approaches, including ecoinvent 3.10 datasets and complementary literature data. Estimates were made over a 100-year time horizon to maintain consistency with the fact that impacts are assessed in the short-term. For incineration, emissions were calculated from the polymer carbon content, distinguishing biogenic and fossil carbon fractions, and crediting the recovered energy according to the substitution-based allocation used in the consequential ecoinvent 3.10 database. For landfilling and composting, degradation rates were taken from Afshar et al.[2], and methane and $CO_2$ emissions were then derived from the assumptions of Benavides et al.[30], which specify the share of polymer carbon degraded, the fraction of methane oxidized, and the resulting atmospheric releases (Supplementary Tables 10 and 11). Degradation rates under marine leakage conditions were based on kinetic data from Corella-Puertas et al.[34], allowing estimation of GHG generation in seawater (Supplementary Table 12). These calculations simplify underlying biochemical processes and thus carry some uncertainty that warrants future refinement.

**Collecting and recycling.** The impacts related to end-of-life plastic collection and recycling were considered burden-free in the assessment of primary plastics (Figs. 1 and 2). For the assessment of production with re-incorporation of recycled materials (Figs. 3 and 4),

impacts were modelled using data from the ecoinvent 3.11 database. Collection and recycling rates were determined based on polymer type and recyclability potential.

### Scenario modeling of large-scale substitution in European packaging

Carbon-reduction trajectories were modeled at the European level, reflecting Europe's central role in the plastics transition[52,53] and the availability of polymer-specific packaging data[5]. Trajectories were based on annual projections of packaging production and demand growth[33] (Supplementary Table 15). Substitution trajectories replacing fossil-based with bio-based packaging followed the configurations of Zheng et al.[46], which consider technical feasibility. *Bio-low* and *Bio-high* scenarios were modeled, corresponding to systems with and without energy decarbonization, respectively. The *Bio-high* scenario uses current electricity mixes from ecoinvent v3.10, while the *Bio-low* scenario applies the decarbonized global mix from SSP2-RCP 1.9, assuming a full renewable transition[54].

For each scenario, polymer-specific material quantities across life cycle stages were calculated from projected European packaging demand using a mass-flow model (Supplementary Fig. 3). The model was parameterized with polymer-specific conversion factors, collection rates, recycling efficiencies, end-of-life pathways, and leakage rates for 2020 and 2050 (Supplementary Tables 1 and 2). Material flows were computed using Supplementary Eqs. (4)–(14). Climate change and ecosystem quality damages were then quantified by combining life cycle stage-specific material quantities with the corresponding impact factors per kilogram of polymer.

The dashed red line in Fig. 4a represents an indicative 1.5 °C-aligned GHG emissions pathway for the European plastics packaging system, derived from IPCC AR6 mitigation scenarios[39]. We operationalized this benchmark by (i) taking the IPCC global emissions trajectory consistent with 1.5 °C, (ii) allocating it to the plastics sector using its share of global GHG emissions reported in the literature, and (iii) scaling to the European packaging scope using the baseline 2020 packaging emissions in our model.

### Impact assessment methodologies

**Methodology.** We used the IMPACT World+ v2.0.1 footprint version in Figs. 1, 3, and 4 to report decision-relevant indicators—climate change, ecosystem quality excluding climate change, human health excluding climate change, water scarcity footprint, and fossil and nuclear energy use—while maintaining methodological consistency with the expert version[55,56]. For climate change, we applied the short-term category using GWP100, which aligns with stakeholder priorities. We also report the "human health (residual)" and "ecosystem quality (residual)" indicators, which exclude contributions from climate change and water availability, as well as all long-term impact categories considered of lower relevance to stakeholders.

Figure 2 presents the contribution of midpoint indicators to the endpoint metrics human health and ecosystem quality, using IMPACT World+ v2.0.1 expert to evaluate environmental damage at the end of the cause–effect chain (including climate change contributions). Only short-term (≤100 years) contributions were retained to ensure consistency with the time horizon considered in the footprint method.

**Land use.** Two types of land use are modeled in the life cycle inventories and assessed following the IMPACT World+ life cycle impact assessment method: land transformation and land occupation. Land transformation (m²) represents the conversion of natural land to a managed use (e.g., deforestation to create cropland). It is a short-term, non-time-dependent flow. Land occupation (m² yr) represents the maintenance of land in a productive state over time (e.g., continued cultivation of cropland). Both flows are recorded at the inventory level and translated into midpoint indicators: occupied area × duration for

land occupation, and transformed area for land transformation. These are subsequently converted to endpoint ecosystem damage using the biodiversity-based CFs developed by de Baan et al.[57], which express the potential loss of species over area and time (PDF m² yr) for each land-use type and ecoregion. These factors express the difference in ecosystem quality between a reference (undisturbed) state and the land-use type considered, based on observed changes in species richness and abundance across multiple taxa.

This framework therefore quantifies direct land-use impacts (occupation and transformation) but does not include indirect land-use change (ILUC), which represents market-mediated effects such as agricultural displacement. ILUC modeling requires economic or equilibrium approaches and is typically addressed within consequential LCA rather than attributional LCA[58].

**Integration of plastics impact assessment.** The environmental damages caused by plastic leakage across the life cycle of bio-based and fossil-based packaging were assessed by coupling inventory-based estimates of plastic losses with the mechanistic CFs from the MarILCA initiative[36], which account for fate, exposure, and effect processes and are conceptually aligned with USEtox[59] (Supplementary Table 13).

The term *leakage* refers to the quantity of plastic that escapes the technosphere and enters the natural environment. Leakage occurs through both losses and releases along various transfer and redistribution pathways. The quantities of plastic losses were estimated using regionally parameterized statistical models, including plastics directly disposed into waterways and oceans, uncollected waste, and collected waste that is poorly managed[60].

At the inventory level, $MP_1$ emissions originate from the leakage of pellets during polymer production, whereas $MP_2$ result from the fragmentation of macroplastics mismanaged at end-of-life[25,32,33]. Plastic losses from other sources, such as transportation or the use phase, were excluded from the study due to their relatively minor contribution and the lack of available data. The generation of $MP_2$ was estimated based on an average fragmentation rate of macroplastics released at end-of-life. A value of 0.5% was adopted, reflecting the expected low fragmentation rate of large macroplastics over a short time horizon (100 years)[34]. Future research should ideally incorporate polymer-specific fragmentation rates to dynamically estimate $MP_2$ formation. $MP_1$ were modelled as spherical particles with a diameter of 1000 μm, while $MP_2$ were represented as spherical particles of 100 μm[34]. Ideally, a statistical estimation of particle sizes and shapes could be performed in future work to assign more realistic physical properties to these fluxes.

At the impact-assessment level, the MarILCA CFs translate these emissions into potential damage to marine ecosystems. The fate component models the distribution and persistence of plastic particles across environmental compartments, including both the water column and marine sediments, and depends on polymer-specific surface loss rates, shape, and size of debris, as degradation primarily occurs at the surface of plastic fragments[35,61]. The exposure–effect component links modeled environmental concentrations in the water column and sediments with experimentally derived effect data[34,62], using a single generic factor that quantifies the physical harm of MPs to marine organisms, as polymer-specific differentiation is not yet available in the literature[36,63]. In the absence of robust toxicity data on the effects of macroplastics on marine biota (e.g., through entanglement), the impacts of macro debris were not included in this study. Similarly, limited data are available regarding the toxicity of polymer degradation products once released into the marine environment. Consequently, the potential effects associated with the leaching of additives could not be incorporated into this assessment.

In IMPACT World+, endpoint indicators such as (eco-)toxicity distinguish short-term (100 years) from long-term (>100 years) impacts, whereas MarILCA CFs integrate emissions over an infinite time horizon. To ensure consistency, we reconstructed finite-horizon fate factors by adapting the MarILCA fate matrix to a 100-year temporal boundary, using degradation and transfer data from Nadim et al. (2025)[36]. The fate model represents all first-order degradation and inter-compartment transfer processes through a multi-compartment rate matrix (K), which governs the time-dependent mass of particles in each environmental compartment. Integrating this coupled system over a finite time horizon allows quantification of the fraction of total mass residence captured within 100 years. This provides short-term fate factors consistent with the IMPACT World+ threshold, while the remaining fraction defines the long-term contribution. Rapidly degrading or mobile plastics are almost entirely accounted for within the short-term horizon, whereas highly persistent polymers retain a significant residual burden beyond it. The analytical derivation of these finite-horizon matrices and their implementation are detailed in Supplementary Fig. 2 and Supplementary Eqs. (1)–(3).

Two contrasting end-of-life scenarios were also modeled to reflect differences in waste-management efficiency (Fig. 2): a high-income context assuming 4% plastic losses to the environment and a low-income context assuming 95% losses.

## Uncertainty analysis

**Inventory data.** A Monte Carlo uncertainty analysis was performed to evaluate the robustness of the results and to account for variability in both foreground and background datasets. The analysis focused on inventory data, since uncertainties in impact assessment models were not included. IMPACT World+ does not provide standard deviations for its characterization factors, except for the plastic-impact category described below.

Uncertainties in inventory data were quantified assuming lognormal distributions. For all background datasets, the geometric standard deviations provided in ecoinvent 3.10 were used directly, based on its built-in pedigree matrix approach[64]. For the foreground systems developed in this study (production and end-of-life treatment of plastics), pedigree scores were established qualitatively according to data source, data generation method (industrial report, experimental data, or modelled estimate), publication date, and geographic relevance. These scores were converted into uncertainty factors following the ecoinvent convention and applied to compute geometric standard deviations for each exchange. The resulting uncertainty ranges served as input for the Monte Carlo simulation (1000 iterations). Due to the presence of both positive and negative balancing flows for freshwater in ecoinvent, the Monte Carlo propagation for water-related categories occasionally yielded non-physical results (e.g., negative net water use). Consequently, uncertainty could not be reliably computed for ecosystem quality and water scarcity indicators, which are therefore reported as deterministic midpoint and endpoint values.

**Plastic impacts.** The uncertainties associated with the CFs (Supplementary Fig. 2) were estimated as 95% confidence intervals using a Monte Carlo simulation with 5000 iterations, based on data provided by Saadi et al.[36].

## Data availability

All primary data used in this study, except for background life cycle inventory data, are available within the Article and its Supplementary Information. The background life cycle inventory data were obtained from the ecoinvent database (https://ecoinvent.org/), available under restricted access due to licensing requirements. All data generated in this study are provided in the Source Data file. Source data are provided with this paper.

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

## Acknowledgements

B.E. acknowledges the financial support provided by the French Agency for Ecological Transition (ADEME) and the Nouvelle-Aquitaine Region.

## Author contributions

B.E. and P.L. designed the scientific approach, analyzed the technical data, and wrote the manuscript. V.C., G.S., and P.L. secured funding and contributed to the manuscript revision.

## Competing interests

The authors declare no competing interests.
