## [Transparent Peer Review file · Nature Communications]

Transition to bio-based plastic packaging reveals complex climate–biodiversity trade-offs

Corresponding Author: Dr Philippe Loubet

Version 0:

Reviewer comments:

Reviewer #1

(Remarks to the Author)

The study presents several noteworthy results. It establishes that bio-based plastics lead to a clear reduction in greenhouse gas (GHG) emissions compared to fossil-based plastics. However, this climate benefit comes at the cost of significantly higher ecosystem damage, primarily due to land use changes and the demands of agricultural input. This trade-off between climate mitigation and biodiversity preservation is a central theme throughout the study. The research also highlights material-specific impacts; such as how corn stover-based PLA offers reduced land-use intensity but incurs higher energy conversion costs. Another important and innovative aspect is the inclusion of marine plastic leakage within the life cycle assessment (LCA), quantifying both micro- and macro plastic impacts—an area often overlooked in traditional LCA studies. Furthermore, through macro-scale modelling of European plastic demand from 2020 to 2050, the authors show that even a full substitution of fossil-based plastics with bio-based alternatives cannot align emissions with IPCC targets unless annual demand is reduced by at least 3%. This demonstrates the critical importance of demand-side interventions alongside material substitution.

The work is significant to the field of sustainability science, industrial ecology, and environmental policy. It offers a comprehensive and systemic multi-scale LCA that incorporates climate, biodiversity, and human health dimensions, as well as plastic leakage—a considerable expansion beyond traditional climate-centric LCAs. Compared to existing literature such as that by Zheng & Suh (2019), Spierling et al. (2018), and Van Roijen & Miller, (2022), this study makes a novel contribution by integrating harmonized inventory datasets across multiple plastic types, modeling scale-up effects at a macro-regional level (Europe), applying advanced methods like Impact World+ and MarILCA, and connecting material-level performance with systemic sustainability narratives. These advances make it a valuable reference for both academic researchers and policymakers navigating the complex trade-offs in circular bioeconomy transitions.

The study's conclusions are well supported by the evidence presented. Quantitative LCA results and scenario modelling (as shown in Figures 1–4) substantiate the claims made. The work demonstrates strength in its detailed modeling of polymer-specific and feedstock-specific impacts, its use of validated databases such as ecoinvent 3.10, and its clear separation of impact categories to avoid double counting. However, the study could be strengthened by including a deeper discussion of third-generation feedstocks (e.g., algae or microbial biomass), which would help balance the current focus on first- and second-generation biomass. Additionally, sensitivity analyses on critical variables like land-use change or plastic leakage rates would enhance the robustness and policy relevance of the conclusions.

There are no critical flaws in the data analysis or interpretation that would prevent publication. However, a few areas warrant clarification or minor revision. For instance, the general conclusion that bio-based plastics increase ecosystem damage could benefit from more nuanced framing, emphasizing that this outcome is particularly tied to first-generation feedstocks and current agricultural practices. Uncertainty quantification is another gap: the figures do not present confidence intervals or variability ranges, and their inclusion would improve transparency. Furthermore, while the marine plastic leakage modeling is cutting-edge, it relies on relatively new frameworks (e.g., MarILCA) that are not yet standardized in LCA practice; this should be more explicitly acknowledged as a limitation.

Methodologically, the study is sound and meets the high standards of industrial ecology and environmental systems research. The use of ImpactWorld+ v2.0.1 provides international comparability, and the harmonization of life cycle inventory data from various sources is carefully executed. The treatment of end-of-life pathways—including landfill, composting, incineration, and recycling—is transparent and well integrated. A key strength is the study's high level of methodological transparency and reproducibility. However, one limitation is that certain biodegradation pathways (e.g., composting or marine environments) are modeled in simplified ways that do not yet include degradation kinetics or additive behavior—future research should aim to address this.

The Methods section provides enough detail for the work to be reproducible. It clearly describes functional units and system boundaries for material-, product-, and macro-level assessments. Procedures for inventory harmonization and classification are well explained, and assumptions regarding carbon sequestration credits for different bio-based polymers are explicitly outlined. Modeling of end-of-life impacts, including plastic leakage and biogenic emissions, is also thoroughly detailed. To further improve clarity, the authors could add a summary table of key parameters and assumptions in the main text and more explicitly state the geographic and temporal boundaries for feedstock production scenarios. Overall, the methodological transparency and attention to reproducibility make this study a reliable and impactful contribution to the field.

Supplementary information

The supplementary information is scientifically robust, methodologically sound. It contains comprehensive dataset supporting the main manuscript, including detailed life cycle inventory data for both conventional and bio-based plastic packaging materials, conversion and end-of-life parameters, and greenhouse gas (GHG) emission models under various waste management scenarios such as landfilling, composting, and marine littering. Furthermore, the inclusion of microplastics-related impacts in the life cycle assessment (LCA) adds significant value, reflecting the authors' engagement with emerging environmental concerns.

However, several issues require attention to ensure clarity and consistency. The document currently contains numerous typographical and formatting inconsistencies, particularly in the tables. Decimal separators are inconsistently applied (e.g., "0.46" instead of "0.46"), and some rows contain misaligned entries or ambiguous abbreviations. Tables such as Supplementary Table 1 and those detailing LCI data (Tables 3–6) are densely packed, which hampers readability. Additionally, Supplementary Figures 1 and 2 are referenced but not clearly presented; high-resolution versions with detailed captions and proper axis labeling should be included. Moreover, key methodological assumptions—such as degradation rates over 100 years or the basis for methane generation and capture—are listed but not sufficiently explained. Providing short narrative justifications or references for these assumptions would enhance transparency and reproducibility. Another area that would benefit from clarification is the characterization of microplastics impacts. The parameters and impact categories (e.g., PDF.m².yr/kg) used in Supplementary Table 10, while highly relevant, are technical and would be better understood if accompanied by brief explanatory notes or a glossary. Overall, while the scientific content is solid and valuable, the presentation and formatting should be carefully revised to meet the standards.

Reviewer #2

(Remarks to the Author)

Overall

The authors conduct a harmonized LCA of fossil- and bio-based plastics. They find that while bioplastics reduce GHG emissions, they also increase ecosystem damage compared to fossil-based plastics, primarily due to land use and unmanaged emissions. The net outcome is found to be sensitive to feedstock origin and waste treatment. When evaluating the results at the scale of Europe, the authors argue that, in part due to these tradeoffs, complete substitution with bio-based plastics cannot offset the impacts of continued plastic demand growth. In closing, the authors advocate for more demand-side measures to mitigate trade-offs.

Overall, this is an interesting paper with useful findings. This field is currently receiving a lot of attention, and more studies that assess tradeoffs, as opposed to simply the impacts of different kinds of materials are needed. The authors incorporate both an assessment of the land impacts of bioplastics, as well as end of life treatments, to reach their conclusions. Given the need for more work to address these critical questions, as well as the authors' thoughtful approach, I recommend this paper for publication.

My main concern with the paper is related to uncertainty, and how uncertainty affects the claims made in the paper. Due to the complexity of the scenarios and uncertainties in what we know, the authors rely on significant assumptions to reach their final conclusion. To some extent, this is unavoidable. Given that this paper is submitted in a multidisciplinary journal and makes claims about plastics policy, however, I would encourage the authors to think beyond their initial findings and provide greater insight into what assumptions underlie their claims. In doing so, showing the reader how their conclusions are derived from the data, the assumptions within that data, and what uncertainties might be overlooked in making these larger claims. I will discuss this in more detail at the end of the review.

Line-by-Line

One minor clarification comment: On line 1 the authors cite a statistic about packaging accounting for 40% of total production, but in line 25, they cite this as both "in EU-28" and "global plastic production". Clarity on what this 40% represents would be appreciated.

Line 32 - The authors mention entanglement and ingestion as a plastic impact, but this paper does not evaluate macroplastic impacts (line 422). Throughout the paper, the authors could be clearer on what is measured and what is not. This needn't go in the introduction, and could go in the methods or supplementary, but given the policy-relevance, more detail on what is not evaluated is warranted.

Line 36 (and throughout the introduction) – Consider specifying what kind of bio-based plastics are being discussed. Drop-in bioplastics have different uses, impacts, and end-of-life than compostable bio-based plastics. Describing these factors and differences between bioplastic types may be useful to the reader.

Line 43 - Similar to the above comment, the authors could also mention composting infrastructure, as not all bio-based plastics would likely enter recycling infrastructure.

Line 46 - Second-generation feedstocks could also have land-use impacts, depending on the system of production. If the feedstock displaces a different land-use, land-use change could still occur no matter the feedstock being utilized. (e.g. <https://doi.org/10.1111/gcbb.70024>)

Line 83-88 - What specifically is being measured to quantify land-use impacts? The authors use “land footprint”, “land occupation”, and “land-use intensity”. Is the measurement simply the total land-use required to produce the target quantity of bio-based plastic? As such, does this exclude potential land-use displacement effects? Given that one of the main trade-offs discussed in this paper is derived from the land-use impacts, greater detail should be provided that discusses 1. How specifically land-use is calculated, and 2. How this value is converted to the final “impact” value.

Line 91 - Similar to the above point, is the decrease in land-use impact from the shift to second-generation feedstocks simply because these are assumed to be residues and not have a land-use impact? What is the assumption about how these second-generation feedstocks will be produced?

Line 96-111 - In evaluating the impact of plastic leakage, if my understanding is correct, the authors rely on several major assumptions about the impacts of plastic waste in the environment, including a degradation model and a microplastic exposure model. These are both areas of study with significant uncertainty, and much of the research is theoretical modeling and not experimental. While this doesn't invalidate this research, greater attention should be given to describing some of these assumptions within the manuscript. Given the potential uncertainties, the authors are potentially propagating significant uncertainty throughout the calculator to arrive at their final estimate. Additionally, because of data limitations, the authors do not include measurement of macroplastics or human toxicity, both of which could be impactful. Again, while this does not detract from the study, a discussion of what is included/not included somewhere in the paper is warranted.

Line 112 - In this paragraph, it seems that direct human toxicity-related impacts, in both the use phase and the end-of-life/leakage phase are not included in this study. Greater clarification on what is and is not included would be appreciated.

Line 218 – This sentence is confusing, as it seemingly relates a $\text{PDF} \cdot \text{m}^2 \cdot \text{yr}$ value to a total amount of arable land. Is this saying the impact would be that significant? Also, while this may seem high, reaching 100% bio-based plastic by 2050 is extremely unlikely. This is a useful heuristic, but the authors should qualify the likelihood of reaching this value, as well as with the fact that a lot of the land production would not occur in the EU.

Line 222 - This paragraph, demonstrating overall demand growth in plastic has a larger impact than the different types of interventions, is useful. The takeaway, that the demand impact is greater than the difference in impact between types of plastic, could be clarified and emphasized more in this section.

Line 243 – The authors claim that ‘bio-based materials offer a promising route to decarbonize the plastics sector’ but is this actually established in the paper? It seems the central argument of the paper is that there are potentially serious tradeoffs for utilizing bio-based plastics as a solution.

Line 256 - Is this the recycling rate for packaging? It seems high for the global level.

Line 328 – What land-use impacts are accounted for? Are emissions from land-use change evaluated?

Finally, two papers that may be useful:
<https://doi.org/10.1038/s41893-024-01492-7>
<https://doi.org/10.1038/s41586-022-05422-5>

Plastic sustainability and plastics policy

The plastic crisis, and how to address it, is a high-profile topic. Because this is a topic with policy-relevance, and the authors make claims about policy goals, the authors should be more forthcoming about the basis on which they are making those claims. This is especially important since this is a multidisciplinary journal, thus not specific to the LCA field and will be read by people with an interest and expertise in the topic, but not necessarily the methodology.

Because LCA studies build on other studies, one potential risk is propagating errors and uncertainties, and thereby affirming assumptions that are already codified in the study.

For example, the authors argue that second-generation feedstocks “reduce the land-use contribution to ecosystem damage” (Line 91), as the example second-generation biomass is a residue, corn stover. But this may not necessarily be the case. A dedicated energy crop that requires agricultural land may require dedicated land for production, especially if it is less land-use efficient than first generation crops. Now, in many circumstances the assumption in the study probably holds. To be more valuable to decisionmakers, however, the authors should be more forthright in describing these assumptions so that it is clearer to the reader under what circumstances the study applies, and does not apply. For example, the authors could specify that corn stover is the example second-generation feedstock, and discuss ranges of potential impact for other feedstocks where data permits. This could go in a supplementary table.

Similarly, the authors could provide some estimates of potential variation in the study. The authors mention different impacts between sugarcane production in Brazil and corn production in the United States. Demonstrating the magnitude of this variation is useful for establishing the magnitude of the impact compared to the uncertainty and range of variation within the scenario.

Relatedly, the authors include microplastic impacts, but exclude macroplastic impacts when discussing impacts of plastic leakage in the environment. It is entirely possible that the impact of what is measured (microplastic impacts) is less than what is not measured (macroplastic impacts). However, because of the uncertainty in this assessment and limited data, this is simply unknown. The authors should be more forthcoming in what is known and not known to again, help the reader in evaluating how this study applies in other contexts. Additionally, when there are estimates with significant uncertainties, such as in the impacts of microplastics on biota and microplastic degradation rates, the authors should acknowledge the uncertainty.

When it is possible to provide measures of uncertainty, or ranges of potential values, I would encourage the authors to do so to demonstrate the size of uncertainty relative to the magnitude of differences between the scenarios.

While true uncertainty estimates may not be possible, I would at the very least encourage the authors to describe more of the assumptions within the study – and key uncertainties therein – so that it is easier to distinguish under what circumstances the assumptions do, and don't, hold. The authors could provide more of this detail in text in the supplementary. Another option would be to create a table of key findings (i.e. for the various categories of impact, what are the largest contributors to the impact) and the key assumptions that underlie those findings. The authors could also consider making Figure 3a and 3b more useful in the life-cycle diagram by illustrating how and where in the plastics lifecycle these impacts arise, thereby showing which impacts are captured and which are not.

Version 1:

Reviewer comments:

Reviewer #1

(Remarks to the Author)

I am satisfied with the revision. The MS can be accepted for the publication.

Reviewer #2

(Remarks to the Author)

The authors have sufficiently addressed my comments on the manuscript. In particular, I appreciate the time and effort that went into the uncertainty analysis and inclusion of greater methodological details.

Point-by-point response to the reviewers' comments

Manuscript ID.: NCOMMS-25-48982A

Reviewer #2:

N°	Reviewer's comments	Authors' responses	Changes in the manuscript
2.1	Line 1: The authors cite a statistic about packaging accounting for 40% of total production, but in line 25, they cite this as both "in EU-28" and "global plastic production". Clarity on what this 40% represents would be appreciated.	The statement has been clarified to specify that the 40% share refers to European plastic production. A supporting reference has been added to substantiate this figure, particularly for the share of packaging used for food applications.	L.9 L.24-25
2.2	Line 32: The authors mention entanglement and ingestion as a plastic impact, but this paper does not evaluate macroplastic impacts (line 422). Throughout the paper, the authors could be clearer on what is measured and what is not. This needn't go in the introduction, and could go in the methods or supplementary, but given the policy-relevance, more detail on what is not evaluated is warranted.	We have specified in the Results section what is actually being evaluated. We have clarified in the Methods section that our assessment focuses exclusively on the physical effects of microplastics on marine biota. Macroplastic-related effects (e.g., entanglement) and additive toxicity are excluded due to the current lack of robust characterization factors.	L.128-133 L.509-516
2.3	Line 36: Consider specifying what kind of bio-based plastics are being discussed. Drop-in bioplastics have different uses, impacts, and end-of-life than compostable bio-based plastics. Describing these factors and differences between bioplastic types may be useful to the reader	We have expanded the introduction to distinguish between the two main categories of bio-based plastics: "drop-in" bio-plastics (e.g., bio-PE, bio-PET) and compostable bio-plastics (e.g., PLA, PHA).	L.36-40
2.4	Line 43: Similar to the above comment, the authors could also mention composting infrastructure, as not all bio-based plastics would likely enter recycling infrastructure	The revised paragraph on bio-based plastic types now explicitly introduces industrial composting as the corresponding end-of-life pathway for compostable plastics, in parallel with recycling for drop-in plastics.	L.46-48

2.5	Line 46: Second-generation feedstocks could also have land-use impacts, depending on the system of production. If the feedstock displaces a different land-use, land-use change could still occur no matter the feedstock being utilized.	We have clarified that the sustainability of second-generation biomass depends strongly on the local production system and residue management practices. The revised text now specifies that indirect land-use change can still occur when the sourcing of non-edible biomass displaces existing land uses or alters soil functions.	L. 52-56
2.6	Line 83-88: What specifically is being measured to quantify land-use impacts? The authors use “land footprint”, “land occupation”, and “land-use intensity”. Is the measurement simply the total land-use required to produce the target quantity of bio-based plastic? As such, does this exclude potential land-use displacement effects? Given that one of the main trade-offs discussed in this paper is derived from the land-use impacts, greater detail should be provided that discusses 1. How specifically land-use is calculated, and 2. How this value is converted to the final “impact” value.	We now clarify that two LCI land-use flows are modeled: land occupation (m ² ·yr) and land transformation (m ²). For each product system we (i) sum these flows to obtain the two midpoint indicators (occupation over time; transformed area), and then (ii) convert them to endpoint ecosystem damage using the Impact World+ characterization factors of de Baan et al. (2013), which link each land-use type and ecoregion to PDF·m ² ·yr (potentially disappeared fraction of species × area × time). This harmonized approach replaces the earlier wording (“land footprint”, “land-use intensity”). Indirect land-use change (ILUC) is not included, as it is a consequential effect requiring market-mediated modeling; this limitation is noted in the Methods section.	L.106-109 L.192-194 L. 471-485
2.7	Line 91: Similar to the above point, is the decrease in land-use impact from the shift to second-generation feedstocks simply because these are assumed to be residues and not have a land-use impact? What is the assumption about how these second-generation feedstocks will be produced?	The lower land-use impacts associated with second-generation feedstocks arise from the economic allocation applied between maize grain and corn stover in the life-cycle inventories. As corn stover has a relatively low market value, a smaller share of the land occupation and transformation burdens is attributed to it, resulting in reduced ecosystem damage per kilogram of polymer. This assumption reflects an attributional perspective, in which residues are treated as co-products of existing agricultural systems. Indirect land-use change (ILUC) effects, which would require consequential modeling of market-mediated land expansion, are not included in this study and are acknowledged as a limitation in the Methods section.	L.115-123

2.8	Line 96-111: In evaluating the impact of plastic leakage, if my understanding is correct, the authors rely on several major assumptions about the impacts of plastic waste in the environment, including a degradation model and a microplastic exposure model. These are both areas of study with significant uncertainty, and much of the research is theoretical modeling and not experimental. While this doesn't invalidate this research, greater attention should be given to describing some of these assumptions within the manuscript. Given the potential uncertainties, the authors are potentially propagating significant uncertainty throughout the calculator to arrive at their final estimate. Additionally, because of data limitations, the authors do not include measurement of macroplastics or human toxicity, both of which could be impactful. Again, while this does not detract from the study, a discussion of what is included/not included somewhere in the paper is warranted.	The reviewer is fully correct that our assessment of plastic leakage relies on several assumptions regarding degradation, fate, and effect models. We have therefore revised and expanded the methodology accordingly. The Methods section now incorporates the latest developments from the MariLCA framework, including updated characterization factors that account for microplastic effects in both the water column and marine sediments, as well as a clearer description of the loss–fate–effect modeling approach. These revisions further distinguish between primary microplastic emissions (from pellet losses) and secondary ones (arising from macroplastic fragmentation), as well as between short- and long-term impacts based on degradation dynamics. Two contrasting high- and low-income country scenarios were further introduced to represent differences in end-of-life mismanagement rates. A dedicated uncertainty analysis was performed using Monte Carlo simulations (5 000 iterations) on the microplastic characterization factors. Finally, we note that human-health effects related to chemical additives or microplastic exposure are not yet included, owing to the current lack of robust characterization factors. This limitation is now explicitly acknowledged in the Methods and Discussion sections.	L.486-531 L.550-552 L.128-146 Supplementary fig. 2 Supplementary table 10
2.9	Line 112: In this paragraph, it seems that direct human toxicity-related impacts, in both the use phase and the end-of-life/leakage phase are not included in this study. Greater clarification on what is and is not included would be appreciate	The reviewer is right. We clarified that direct human exposure to microplastics or chemical additives during the use and leakage phases is not included in conventional attributional LCAs, as no validated human-health fate–exposure–effect characterization models are currently available for plastics.	L.149-152
2.10	Line 218: This sentence is confusing, as it seemingly relates a PDF*m2*yr value to a total amount of arable land. Is this saying the impact would be that significant? Also, while this may	The PDF·m²·yr metric was clarified as an endpoint indicator representing the total potential loss of species from all contributing environmental stressors. We also specified that land-use pressures (occupation and	L. 263-272

	seem high, reaching 100% bio-based plastic by 2050 is extremely unlikely. This is a useful heuristic, but the authors should qualify the likelihood of reaching this value, as well as with the fact that a lot of the land production would not occur in the EU.	transformation) are the main drivers of this aggregated biodiversity damage. The comparison to European arable land was retained as an illustrative benchmark to convey magnitude, while acknowledging the scenario's limited feasibility and global production scope.	
2.11	Line 222: This paragraph, demonstrating overall demand growth in plastic has a larger impact than the different types of interventions, is useful. The takeaway, that the demand impact is greater than the difference in impact between types of plastic, could be clarified and emphasized more in this section.	A paragraph was added to clarify that demand growth remains the dominant driver of environmental impacts and to emphasize that substitution strategies must be coordinated with broader demand-side policies and material sufficiency measures.	L. 280-282
2.12	Line 243: The authors claim that "bio-based materials offer a promising route to decarbonize the plastics sector" but is this actually established in the paper? It seems the central argument of the paper is that there are potentially serious tradeoffs for utilizing bio-based plastics as a solution.	The paragraph has been reformulated to better reflect the findings. It now explicitly acknowledges that while bio-based materials can contribute to decarbonization, this comes with significant trade-offs in terms of ecosystem quality, thereby clarifying that substitution alone cannot ensure sustainable transitions.	L. 295-297
2.13	Line 256: Is this the recycling rate for packaging? It seems high for the global level	The recycling rate values (23% and 94%) are taken from Bachmann et al. (2023). The lower value represents the current global average recycling rate for plastic packaging, while the upper value corresponds to a theoretical technical maximum derived from material flow optimization under ideal collection and processing conditions. These values are used to illustrate the potential mitigation range rather than to represent observed global recycling rates.	L.314
2.14	Line 328: What land-use impacts are accounted for? Are emissions from land-use change evaluated?	As clarified earlier (response to comment 2.1), indirect land-use change (ILUC) is not included in this study. ILUC represents a consequential, market-mediated effect that falls outside the scope of attributional LCA. Only direct land occupation and transformation are modeled, as detailed in the Methods section. This limitation is explicitly acknowledged in the Methods.	L.471-485

2.15	Finally, two papers that may be useful: https://doi.org/10.1038/s41893-024-01492-7 https://doi.org/10.1038/s41586-022-05422-5	We thank the reviewer for recommending these two references. Although we were aware of their existence and had built part of our rationale upon their findings, they had been omitted from our citation list. We have therefore slightly restructured the introduction to include them and to clarify our positions. We have also improved the presentation of the study's objectives, scope, and limitations.	L.59-80
2.16	The authors argue that second-generation feedstocks “reduce the land-use contribution to ecosystem damage” (Line 91), as the example second-generation biomass is a residue, corn stover. But this may not necessarily be the case. A dedicated energy crop that requires agricultural land may require dedicated land for production, especially if it is less land-use efficient than first generation crops. Now, in many circumstances the assumption in the study probably holds. To be more valuable to decisionmakers, however, the authors should be more forthright in describing these assumptions so that it is clearer to the reader under what circumstances the study applies, and does not apply. For example, the authors could specify that corn stover is the example second-generation feedstock, and discuss ranges of potential impact for other feedstocks where data permits. This could go in a supplementary table.	We understand the reviewer's concern, which relates to potential land-use competition arising when second-generation feedstocks are sourced from dedicated energy crops rather than residues. In this study, second-generation biomass is represented by corn stover, which is modeled as an agricultural residue and therefore does not induce additional land occupation. Indirect land-use change (ILUC) effects from dedicated crops are not included, as they fall under consequential modeling beyond the scope of this attributional LCA. This limitation and its implications are now explicitly mentioned in the Methods.	L.106-109 L.471-485
2.17	Similarly, the authors could provide some estimates of potential variation in the study. The authors mention different impacts between sugarcane production in Brazil and corn production in the United States. Demonstrating the magnitude of this variation is useful for establishing the magnitude of the impact	We thank the reviewer for this valuable suggestion. To better capture variability in production conditions, Figure 3 has been revised. Panel c now focuses on climate change and ecosystem quality impacts, allowing the introduction of two new scenarios (PLA-low and PLA-high) that explicitly represent the range of possible outcomes for agricultural location and energy decarbonization.	L.211-234

	compared to the uncertainty and range of variation within the scenario.	These scenarios were designed to reflect the differences in impact observed between sugarcane production in Brazil and corn production in the United States, as mentioned in the text, thereby illustrating the potential variation within realistic global production conditions. The results show that while production geography and energy intensity affect total impacts, the feedstock type remains the dominant driver, with larger differences between first- and second-generation feedstocks than between regional settings. This addition provides a transparent quantification of variability and situates it relative to the underlying uncertainty of the results.	
2.18	Relatedly, the authors include microplastic impacts, but exclude macroplastic impacts when discussing impacts of plastic leakage in the environment. It is entirely possible that the impact of what is measured (microplastic impacts) is less than what is not measured (macroplastic impacts). However, because of the uncertainty in this assessment and limited data, this is simply unknown. The authors should be more forthcoming in what is known and not known to again, help the reader in evaluating how this study applies in other contexts. Additionally, when there are estimates with significant uncertainties, such as in the impacts of microplastics on biota and microplastic degradation rates, the authors should acknowledge the uncertainty.	Please refer to comment 2.8	L.486-531 L.550-552 L.128-146 Supplementary fig. 2 Supplementary table 10
2.19	When it is possible to provide measures of uncertainty, or ranges of potential values, I would encourage the authors to do so to demonstrate the size of uncertainty relative to the magnitude of differences between the scenarios.	We agree with the reviewer. To quantify the magnitude of uncertainty relative to inter-scenario differences, we performed Monte Carlo simulations (1,000 iterations) to propagate uncertainty in inventory data for polymer production. The 5th–95th percentile ranges are now displayed as error bars in Figure 1. Uncertainties were derived from the ecoinvent 3.10	L.86-90 L.533-549

		pedigree matrix, complemented by quality assessments of the foreground agricultural and industrial datasets compiled in this study. This analysis captures the variability associated with life cycle inventory data only, as standard deviations are not provided for current LCIA characterization factors. Because of the numerical instability introduced by freshwater flows inecoinvent (which can alternate between positive and negative values during Monte Carlo sampling), uncertainty ranges could not be robustly computed for Ecosystem Quality and Water Scarcity. These categories are therefore reported as deterministic averages to ensure consistency. This limitation is noted in the Methods section.	
2.20	While true uncertainty estimates may not be possible, I would at the very least encourage the authors to describe more of the assumptions within the study – and key uncertainties therein – so that it is easier to distinguish under what circumstances the assumptions do, and don't, hold. The authors could provide more of this detail in text in the supplementary. Another option would be to create a table of key findings (i.e. for the various categories of impact, what are the largest contributors to the impact) and the key assumptions that underlie those findings. The authors could also consider making Figure 3a and 3b more useful in the life-cycle diagram by illustrating how and where in the plastics lifecycle these impacts arise, thereby showing which impacts are captured and which are not.	We thank the reviewer for this constructive suggestion. We have strengthened the discussion of uncertainty and assumptions throughout the revised manuscript. Figure 1 now explicitly represents uncertainty ranges derived from the Monte Carlo analysis of inventory data, while Figure 3 explores scenario variability across different feedstock, and end-of-life conditions. Together, these elements aim to clarify how model assumptions influence the results and their robustness. If the reviewer and editor consider it useful, we would be glad to include in a subsequent revision a supplementary table summarizing the key assumptions and dominant contributors for each impact category.	

Reviewer #1:

N°	Reviewer's comment	Authors' response	Changes in the manuscript
1.1	The study could be strengthened by including a deeper discussion of third-generation feedstocks (e.g., algae or microbial biomass), which would help balance the current focus on first- and second-generation biomass.	We agree that third-generation feedstocks, such as algal or microbial biomass, represent an emerging and promising pathway for bio- based plastic production. But given this limited data availability and the current uncertainty surrounding their large-scale deployment, we chose to focus our assessment on first- and second-generation feedstocks, for which more robust and comparable datasets exist. We have clarified the scope in the revised manuscript.	L.77-80 L.300-307
1.2	Additionally, sensitivity analyses on critical variables like land-use change or plastic leakage rates would enhance the robustness and policy relevance of the conclusions.	We agree with the reviewer on the relevance of conducting deeper sensitivity analyses. Figure 3 has been revised to include a sensitivity analysis on the geographical location of agricultural production phases. Please refer to comment 2.17 for further details. Regarding plastic leakage, additional loss scenarios have also been incorporated into Figure 2. Two contrasting cases representing high- and low-income countries were introduced to reflect differences in end-of-life mismanagement rates.	L.210-234 L.141-146
1.3	However, a few areas warrant clarification or minor revision. For instance, the general conclusion that bio-based plastics increase ecosystem damage could benefit from more nuanced framing, emphasizing that this outcome is particularly tied to first-generation feedstocks and current agricultural practices.	We have slightly revised the conclusion paragraph to better emphasize the differences between feedstock generations and their respective environmental implications.	L.300-307
1.4	Uncertainty quantification is another gap: the figures do not present confidence intervals or variability ranges, and their inclusion would improve transparency	We added 5th–95th percentile error bars in Fig. 1 from a 1,000-draw Monte Carlo on inventory data, improving transparency on variability (Please refer to comment 2.19)	L.86-90 L.533-549

1.5	Furthermore, while the marine plastic leakage modeling is cutting-edge, it relies on relatively new frameworks (e.g., MarLLCA) that are not yet standardized in LCA practice; this should be more explicitly acknowledged as a limitation.	We now explicitly acknowledge that MarLLCA is an emerging framework in LCA practice and we flag this as a limitation where plastic-leakage impacts are modeled (Please refer to comment 2.8)	L.486-531 L.550-552 L.129-147 Supplementary fig. 2 Supplementary table 10
1.6	However, one limitation is that certain biodegradation pathways (e.g., composting or marine environments) are modeled in simplified ways that do not yet include degradation kinetics or additive behavior—future research should aim to address this.	We fully agree with the reviewer on the importance of better integrating the degradation dynamics of biodegradable plastics at end of life. Although kinetic data for landfill and composting conditions have not yet been implemented, we refined the biodegradation percentages based on a new source that differentiates rates by polymer type (Afshar et al., 2024). The degradation of debris reaching the marine environment was also updated using kinetic data reported by Corella-Puertas et al. 2023. Corresponding GHG emission from these biodegradation processes were revised accordingly, leading to adjustments in the results presented in Figure 3. We have provided a more detailed description of the underlying assumptions in the Methods section.	L.420-432 Supplementary table 7-9
1.7	To further improve clarity, the authors could add a summary table of key parameters and assumptions in the main text and more explicitly state the geographic and temporal boundaries for feedstock production scenarios.	We clarified geographic boundaries for the feedstock production (for PLA scenarios) in Supplementary Table 11 and explained the assumptions in the Fig. 3 caption. If helpful, we can add a concise assumptions summary table to the main text in a subsequent round. (please refer to comment 2.20)	Supplementary table 11
1.8	However, several issues require attention to ensure clarity and consistency. The document currently contains numerous typographical and formatting inconsistencies, particularly in the tables. Decimal separators are inconsistently	We thank the reviewer for noting these formatting inconsistencies, which have now been corrected. The tables have been reorganized for improved readability, and typographical and formatting errors have been addressed.	Supplementary information

	applied (e.g., “0,.46” instead of “0.46”), and some rows contain misaligned entries or ambiguous abbreviations. Tables such as Supplementary Table 1 and those detailing LCI data (Tables 3–6) are densely packed, which hampers readability.		
1.9	Additionally, Supplementary Figures 1 and 2 are referenced but not clearly presented; high-resolution versions with detailed captions and proper axis labeling should be included.	The resolution of the graphs in the main text has been generally improved wherever possible. In the Supplementary Information, figures and tables have been reorganized and accompanied by more detailed captions where necessary.	All figures
1.10	Moreover, key methodological assumptions—such as degradation rates over 100 years or the basis for methane generation and capture—are listed but not sufficiently explained. Providing short narrative justifications or references for these assumptions would enhance transparency and reproducibility.	We agree that key modeling assumptions such as degradation rates and methane generation needed clearer justification. The revised Methods (GHG emissions from incineration...) section now briefly explains the basis and references for these parameters, and Supplementary Tables 7–9 indicate the data sources used for landfill, composting, and marine degradation models.	L.420-432 Supplementary table 7-9
1.11	Another area that would benefit from clarification is the characterization of microplastics impacts. The parameters and impact categories (e.g., PDF.m ² .yr/kg) used in Supplementary Table 10, while highly relevant, are technical and would be better understood if accompanied by brief explanatory notes or a glossary.	To improve the interpretation of the parameters presented in Supplementary Table 10, a brief description has been added below the table, specifying in particular the meaning of the impact units used. More broadly, the general methodology for assessing plastic pollution has been further detailed in the Methods section.	Supplementary table 10